# Amoebae in Chronic, Polymicrobial Endodontic Infections Are Associated with Altered Microbial Communities of Increased Virulence

**DOI:** 10.3390/jcm9113700

**Published:** 2020-11-18

**Authors:** Garrit Koller, Federico Foschi, Philip Mitchell, Elizabeth Witherden, Kenneth Bruce, Francesco Mannocci

**Affiliations:** 1Department of Endodontics, Faculty of Dentistry, Oral & Craniofacial Sciences, Floor 22 Tower Wing, Guy’s Dental Hospital, London SE1 9RT, UK; garrit.koller@kcl.ac.uk (G.K.); pjc.mitchell@virgin.net (P.M.); francesco.mannocci@kcl.ac.uk (F.M.); 2Centre for Host Microbiome Interactions, King’s College London Dental Institute at Guy’s Hospital, King’s Health Partners, London SE1 9RT, UK; elizabeth.witherden@kcl.ac.uk; 3LCN—London Centre for Nanotechnology, 19 Gordon St, Bloomsbury, London WC1H 0AH, UK; 4Centre for Oral, Clinical and Translational Sciences, Floor 25, Tower Wing, Faculty of Dentistry, Oral & Craniofacial Sciences, King’s College London, London SE1 9RT, UK; 5Department of Therapeutic Dentistry, I. M. Sechenov First Moscow State Medical University, 119146 Moscow, Russia; 6King’s College London, Institute of Pharmaceutical Science, Franklin-Wilkins Building, London SE1 9NN, UK; kenneth.bruce@kcl.ac.uk

**Keywords:** endodontics, infection, polymicrobial infection, microbial ecology, microbiology, bacterial fitness, amoeba-resistant bacteria

## Abstract

Background: Infections of the root canal space involve polymicrobial biofilms and lead to chronic, low grade inflammatory responses arising from the seeding of microbes and by-products. Acute exacerbation and/or disseminating infections occur when established microbial communities undergo sudden changes in phenotypic behaviour. Methods: Within clinical endodontic infections, we assessedcategorical determinants comprising, and changing microbial composition of, chronic polymicrobial infections and their association with amoebae. After standardised assessment, primary or secondary infections underwent sampling and DNA processing, targeting bacteria, fungi and amoebae, including 16S high-throughput sequencing. After taxonomic assignment, community composition was correlated with clinical signs and symptoms. Diversity and abundance analyses were carried out in relation to the presence of non-bacterial amplicons. Results: Clinical specimens revealed two distinct community clusters, where specific changes correlated with clinical signs. An association between the compositions of microbiomes was found between these groups and the presence of *Entamoeba gingivalis* in 44% of cases. When amoebae were present in endodontic infections, we demonstrate changes in microbial community structure that mirror those observed in treatment-resistant or recurrent infections. Conclusions: Amoeba are present in endodontic infections at a high prevalence, and may promote increased virulence by enrichment for phagocytosis-resistant bacteria.

## 1. Introduction

Culture-dependent and independent studies have so far demonstrated 800 distinct taxa of bacteria alone, derived from 13 phyla within the oral cavity, with 100–200 thereof typically present in individuals [1,2,3]. Much remains to be characterised, with many bacterial and non-bacterial taxa not having been systematically described and cultured. Within the endodontic infection niche, nine predominant phyla phyla, namely Firmicutes, Bacteroidetes, Actinobacteria, Proteobacteria, Fusobacteria, Spirochaetes, Synergistetes, Candidatus Saccharibacteria (formerly *TM7*) and *Sulfur River 1 (SR1)* are found within complex and diverse communities. More recently, Anderson and co-workers further described phyla *Chloroflexi*, *OD1*, *Verrucomicrobia* and shifts in contributing phyla between asymptomatic and symptomatic teeth. Some authors have attempted to correlate microbial taxa with clinical signs and symptoms of endodontic disease, but no one single microbial species have been identified as ‘keystone’ species to date. Association-based studies have demonstrated links with only few species, such as *Enterococcus faecalis* [1]. In recent years, with advances in molecular, high throughput technologies, the concept of interchangeable, multispecies, and sometimes multi-kingdom, pathogenic diseases have been proposed [4], with the microbiome acting as a pathogenic entity with interchangeable, individual members [5]. Dental infections (of otherwise sterile sites) undergo such transitions to a more symptomatic presentation, including local and systemic dissemination and resistance to treatment, such as chemo-mechanical debridement of the endodontic space. 

In nature, one of the significant influences on bacteria are amoebae, present in a diverse range of ecological niches where nutrients may be scarce and polymicrobial biofilms evolve [6,7,8]. These inter-kingdom interactions have a long evolutionary history of co-existence (exposure) and grazing (predation), with intense selective pressure exerted on bacteria to evade or limit phagocytosis by protists. The presence of protists has been demonstrated to dramatically alter both number and taxonomic distribution of bacteria [9,10,11,12]. Furthermore, amoebae have been shown to selectively package certain amoeba-resistant bacteria (ARB), conferring protection from a hostile environment and chemical disinfectants [13], including the most commonly used disinfectant in endodontology, hypochlorite [14].

Modelled on these observations within such competitive environments, we sought to explore the presence, association and potential impact of amoebae on human polymicrobial endodontic bacterial composition and how these might affect signs and symptoms. We examined the co-existence of associated amoeba with specific emphasis on *Entamoeba gingivalis* (first described by Gros in 1849), associated with periodontal disease severity [15,16] and shown to affect host immune cells directly through a process involving injury and ingestion of host neutrophils and lymphocytic cell nuclei [16], likely due to to a recently-discovered mechanism for cell damage by ingestion of live host cell fragments in a nibbling-like process, akin to trogocytosis in the related species *E. histolyticum* [17].

In the present study, we hypothesised that amoeba-bacterial interactions may correlate, and mechanistically contribute to, microbial communities within endodontic infections and sudden changes to their interaction, including virulence traits consistent with phagocytosis-resistance of bacteria exposed to amoeba.

## 2. Materials and Methods

### 2.1. Sample Collection and Preparation

Twenty-five patients referred to the Endodontic Unit of the Department of Restorative Dentistry, King’s College London, Guy’s Hospital, London, UK, for endodontic treatment were enrolled in this study. The study was approved by the Ethics Committee of King’s College London (05/Q0705/051) Patients over 18 years were recruited following referral to a secondary care hospital and written, informed consent was gained. All research was conducted in accordance with relevant guidelines and complied with the Declaration of Helsinki. Operative procedures were undertaken by endodontic residents, and methods of dental assessment included a full oral examination, periodontal probing to exclude periodontitis and examination, periapical radiographs, palpation and percussion tests, along with recording signs and symptoms including presence of pain, abscess, sinus tract as a measure of chronic suppurative infections, and abnormal mobility as measure of destruction of the supporting bone and periodontal ligament. Clinical metadata collected, sampling methods and details of molecular analyses are provided in Appendix A.

### 2.2. Correlation and Statistical Analysis

#### 2.2.1. Community Clustering

Initial analysis was carried out from phylogenetic assignments, as well as OTU tables to determine the species diversity across all samples and clustering of communities through MNDS unsupervised principal component analyses of reads, expressed as percentage reads to control for read numbers, prior to sub-clustering along with metadata. Correlations were carried with findings of positive 18S amplification, pan-amoebae and Entamoeba status.

Group comparisons were conducted using the STAMP software (http://kiwi.cs.dal.ca/Software/STAMP) (Beiko Lab, Halifax, NS, Canada) bioinformatics and R-based MicrobiomeAnalyst suite (http://www.microbiomeanalyst.ca, [18]). R Studio programs phyloseq, using non-metric multidimensional scaling (NMDS) assessed clustering heterogeneity with Shannon-Jenson algorithms. Heatmaps with Ward-clustered Dendrograms were constructed, and secondary enrichment analyses carried out using Linear Discriminant Analysis Effect Size (LEfSe;) as well as contruction of Feature-level dendrograms to assess Jenson-Shannon divergence for phylogenetic relationships (based on the R-based package hclust) for classes for bivariate data.

#### 2.2.2. Statistical Methods

To evaluate two-tailed correlations between percentage reads of bacteria, as well as the metadata, binned into binary classes for infection types, gender, age and lesion size intervals, as well as Ct values, fungal and amoebal presence, an unsupervised Pearson’s product moment correlation was carried out (SPSS v24, IBM, Chicago, IL, USA).

Pairwise comparisons of binary data were carried out using the Fisher’s exact test. After adjusting for power to account for Type I and II errors, all data samples presented were present >6 cases and significance set at *p* > 0.05/ R > 0.35 or R < −0.17 for correlations). 

Univariate analyses were statistically assessed subjected to a Kruskal-Wallis non-parametric test, followed by posthoc testing for enrichment. Species diversity and richness was computed using EstimateS with 10,000 iterations.

## 3. Results

### 3.1. Patient Cohort

Patient characteristics and demographics for the 25 cases enrolled into this study, derived from one tooth per subject, are provided in Table 1.

### 3.2. Bioinformatic Analysis of the Bacterial Microbiome in Endodontic Lesions

Amplicon sequencing of the V3–V5 hypervariable region of the 16S rRNA gene lead to successful amplification from all samples, with a minimum number of reads set at 30,000. Procedural contamination was ruled out from mock and surface samples, as well as non-template controls. 89% of reads passed initial quality check for quality and length. Operational taxonomic units (OTU) were assigned to the taxa found, and this table provided in Appendix A.

### 3.3. Phylogenetic Grouping of Endodontic Infections

In order to correlate different contributors to endodontic infections, we first characterised the bacterial community across cases. 154 OTUs were identified across all samples at *n* > 2 and >0.01% abundance. Taxonomic assignment to the family and genus level was achieved for 94.5% (Standard deviation 12.7) and 77% (S.D. 25.7) of reads to family and genus levels, respectively, whilst only 26.1% (S.D. 17.1) resolved to species level. 1.19% (S.D. 1.2) of reads could not be assigned to any known taxon, coin keeping with previous reports [3]. A total of 13 phyla were identified. Consistent with previous findings [3,19], the predominant phylogroups by abundance within endodontic infections were *Firmicutes* (constituting 48% of reads, present in 100% of lesions examined at relative abundance of >0.01%), Bacteroidetes (17% of reads, 100% presence), *Proteobacteria* (17% of reads, 100% presence), *Actinobacteria* (13% of reads, 100% presence), *Fusobacteria* (3% of reads, 96%), which defined the core microbiome across all samples. *Candidate phylum Saccharibacterium* (formerly TM7) (80% presence), *Synergistetes* and *Spirochaetes* were present at low relative abundance within samples, but were both present in 76% of samples overall. *Candidate Phylum* Sulphur River 1 (SR1) (36%) and *Tenericutes* (28%) were present at low abundance within and across samples, whilst *Verrucomicrobia* and *Cyanobacteria* reads were found in three samples (12%) and Chloroflexi in only one case.The predominant taxa assigned are given in Figure 1 and Table 2.

Considering the less abundant *phyla* identified, SR1 (OTU 143) could not be subclassified beyond *phylum* level, and the sole members of the phylum Spirochaetes were members of the *genus Treponema sp.*, namely *T. amylovorum* (OTU145), *T. socranskii* (OTU146) and another, unclassified member (OTU144). The sole family represented within the *phylum Synergistetes* was *Desulfovibrionaceae*, comprising members of the genera *TG5* and *Jonquetella* (*J. anthropi*) and another, unresolved genus. Within the epibiont phylum *Candidatus Saccharibacterium* (formerly TM7), three families from class TM7-3 were identified (OTU150-153). Two of these could be assigned within distinct *genera* related to oral clones I025_RS-045 (OTU150) and CW040_/CW040_F16 (OTU151&152). Withineach phyla of *Chloroflexi*, *Verrucomicrobia*, *Tenericutes*, only one taxa each was identified, comprising *SHD-231* (OTU043), *Optitutus sp.* (OTU155) and *Mycoplasma sp.* (OTU154), respectively.

### 3.4. Cluster Analysis of Bacteria Present in Endodontic Lesions

Endodontic infections could be classified into community clusters, covering > 95% of samples at OTU and taxa level. This resulted in two major, categorical clusters with small number of OTUs sharing positive association with both of the two major clusters observed (Figure 2). Some taxa were present across both clusters, without respective group (sub-cluster) correlates, such as OTU079 *Catonella sp.*, OTU107 *Fusobacterium sp.*, OTU029 *Porphyromonas endodontalis,* OTU146 *Treponema socranskii*, OTU150 *TM7-3 or* OTU123 *Neisseria sp.*2 were present in both predominant clusters observed, suggesting niche adaptation, interchangeability of species and functions within both clusters. When examined together, both the of correlates and clusters at different taxonomic levels revealed that orders and families were present consistently, such as the correlation between *Porphyromonaceae*, *Neisseriaceae* and *Veillonellaceae sp.*, but these were lost at lower phylogenetic levels and highlighted that lower taxonomic members could substitute(Figure 2, Figure 3 and Figure 4). This suggested conserved, non-interchangeable interactions and roles of members of individual families and *genera* were observed, as well as the exclusivity of niches such as *Lactobacillus sp.* depauperate states correlating with increased abundance of *Fusobacteria sp* and *Neisseria sp.* Lower level correlations, including OTU-level correlations, are provided in Appendix A.

The OTU correlation of the epibiont and parasitic *phylum* and phylotype TM7 was associated with differential specificity at every phylogenetic level and clusters; TM7 F16 (OTU152) aligned with *Gemellaceae*, *Carnobacteriaceae*, *Micrococcaceae* and *Coriobacteriaceae*, *Neisseriaceae*; *Neisseria cinerea*, *Rothia dentocariosa*, *R. aeria* and *R.mucilaginosa*, the pathogenicity-associated TM7 clonotype RS-045/I025 (OTU153) with *Clostridiaceae*, *Pseudomonaceae*; *Porphyromonas endodontalis*, *Prevotellales (P. tannerae and P. nigrescens)*, as well as *Solobacterium moorei.* CW040 clustered with *Veillonellaceae (V. dispar and V. pallens)* and *Bifidobacteriaceae.*

In order to assess association, between sample types and metadata, the latter were analysed in relation to the reads attained, after normalisation to relative percentages of reads and log normalised reads to account for any large abundance differences. Data resolved for combinedinfection type, age, gender, lesion size, amoebal presence and modifying bacteria selected from the initial correlation analysisare provided in Figure 3 and Figure 4 and Appendix A.

### 3.5. Association with Primary or Secondary Infections

Whilst the abundances of phyla and core microbiome composition for primary and secondary treatment cases shared many members, and shifts in the distribution of predominant *phyla* were identified between treatment types, for *Proteobacteria* (17 and 14% for primary and secondary cases, respectively), *Actinobacteria* (12 and 17%). *Fusobacteria* were enriched 2.5-fold in secondary cases (from 2 to 5%). In contrast, *Synergistetes* was found to be dramatically decreased in secondary cases (2 to <0.1%), as seen in Appendix A, with clustering at genus level demonstrating tighter clustering of secondary cases (Figure 1 and Appendix A Secondary cases displayed a reduced heterogeneity of *genera* present with predominance of taxa *Streptococcus sp.*, *Prevotella sp* and *Rothia sp.*, and some primary cases (P5, P14) resembling the communities observed in secondary infections.

Non-metric, multidimensional scaling was carried out for infection type, signs and symptoms, along with the amoebal amplification (Figure 5 and Figure 6). The beta diversity plotted in samples found to amplify *E. gingivalis* aligned with clusters predominated by secondary infections and those presenting with sinus tracts. The NMDS analysis after distancing of *genera* by Jensen-Shannon Divergence and Jaccard Indices, followed by an analysis of Homogeneity of Group dispersions aligned the amoebal cluster to this (PermDisp, F = 5.821, *p* = 0.028).

### 3.6. Quantitative Polymerase Chain Reaction (qPCR)

The qPCR examining the 16S rRNA gene amplified from all samples examined successfully. The fungal ITS1/2 primer set amplified at Ct values > 2 below control in four of 25 samples, yielding *Candida sp.*/*Saccharomyces sp.* in these cases. Amoebal PCR produced amplicons correlated with *Entamoeba gingivalis* specific amplicons. The PCR based approach showed that 11/25 (44%) samples amplified *E. gingivalis.* Agarose gel electrophoresis and Sanger sequencing confirmed the specific amplification, resulting in two distinct genotypes.

### 3.7. Microbiome Associated with Entamoeba Status

At OTU level, in addition to the shared diversity with secondary cases, significant positive associations were established in *E. gingivalis*-positive samples for the families Microbacteriaceae (OTU008 *p* = 0.003; OTU 007 *p* = 0.0012), Enterococcaceae (OTU061), genus *Methylobacterium* (OTU111, *p* = 0.0025), Genera and species *Actinomyces sp.* (OTU003), *Haemophilus sp.* (OTU136), *Prevotella pallens* (OTU036), *Solobacterium (Bulleidia) moorei* (OTU104), *Neisseria sp.*(OTU122), as well as *Neisseria subflava* (*p* = 0.042), *Porphyromonas endodontalis* (OTU029), *Veillonella sp.* (OTU093, *p* = 0.031), independent of the OTU resolved to species level, of *V. dispar* (*p* = 0.028) and *V. parvula* (*p* = 0.049). *Leptotrichia sp.* (OTU109), *Rothia aeria* (OTU010), *Aggregatibacter segnis* (OTU135, *p* = 0.036), *Prevotella nanceiensis* (OTU034), *Streptococcus sp.* (OTU068) and *Paludibacter sp.* (OTU027), as well as *Rothia sp.* (*p* = 0.012), including *Rothia mucilaginosa* (*p* = 0.039), were identified as enriched. Two members of the genus *Prevotella* were found, namely *P. pallens* (*p* = 0.012) and *P. melaninogenica* (*p* = 0.031), which was one of the predominant OTUs for all samples.

The associated bacterial 16S copy number was 4.4 × 10^5^ and 2.1 × 10^6^ for *E. gingivalis* positive and negative samples, respectively, but no statistical significance could be determined(*p* = 0.26). Concerning clinical parameters, the presence of *E. gingivalis* correlated with pain in 81.8 and 57%, tenderness to percussion in 72 and 42.8%.

An Alpha diversity of 10.98(SD = 0.33) and 11.71 (SD = 0.35), Simpson Inverse Mean of 10.75 and 20.26, Shannon Exponential Mean of 23.93 and 35.13 was observed for the *E. gingivalis* positive group and negative groups, respectively, thus supporting a categorical association between the samples diversity and entamoeba status.

## 4. Discussion

In the natural environment, *amoebae* are significant modifiers of bacterial communities, as well as dramatically altering both microbial number and taxonomic distribution. With the present study, we sought to explore how such shifts may mirror those seen in endodontic infections, symptoms and signs. Whilst several studies have examined the bacterial composition and diversity within the endodontic niche, fewer reports have examined the role that eukaryotic organisms play in modifying the bacterial composition of such infections or their potential contribution to periradicucular translocation. The current study is the first describing associations between polymicrobial communities and the presence of a protist predator within infected root canal spaces of non-vital, immunologically inaccessible teeth. The communities of microbiota encountered within the group described here, agreewith previously published reports of microbial diversity and phylogroups associated with endodontic lesions and align with associations with signs and symptoms [3]. Whilst no mechanistic role could be directly inferred from our data, many organisms associated with these changes were also characterised as emerging, and phagocytosis-resistant pathogens.

This multivariate analysis enabled us to infer a phylogenetic and physiological niche, categorisation into core and keystone phylotypes, as well an indication towards ecological relationships (positive correlations of clusters), in terms of competition (negative correlations), potential parasitism (as seen by the enrichment in epibionts derived from the Phyla *Candidatus Saccharibacterium* or *Tenericutes* (*Mycoplasma sp.*), commensalism by core microbiomes, mutualism and predation by protists.

By examining the differential microbiomes between initial infections and those where treatment failed (secondary cases), suggested communities linked to refractory endodontic disease, as these aligned into a more condensed group of taxa and species found outside the root canal space [20]. The presence of sinus tracts, as a surrogate marker of chronicity of infection [21], collimated with clusters of secondary infections and those found to harbour *E. gingivalis.* Significant changes in composition in less abundant *phyla*, with enrichment in *Actinomyces* and *Fusobacteria* were observed, and a reduction of the prevalence of *Proteobacteria* and *Synergistetes*. The finding of enrichment of *Actinobacteria* and *Fusobacteria* is in agreement with FISH-hybridisation studies for secondary infections [22,23]. Unique and specific correlation between genera, specifically *Porphyromonas, Neisseria* and *Prevotella* species, may suggest predominantly non-interchangeable cooperation. This points to the potential role of *E. gingivalis* in modulating human disease arising from polymicrobial infections, with significant enrichment of bacteria known to cause systemic disease [24,25]. Whilst the study design precludes demonstration of association between the presence of *E.gingivalis* and the onset of diversity changes, the presence of the protist did correlate with a microbiota closely resembling secondary infections. This suggested that a stable, consistent community was attained in failed cases and when *E.gingivalis* was present. This suggests a functional role in selecting for the more condensed and pathogenic community seen in persisting/recurrent infections, possibly caused, or contributed to, by amoeba. Concerning other eukaryotic species present with lesions, and a shortcoming of the present study, these were underpowered in terms of abundance to ascribe correlations or significance.

The presence of *E. gingivalis* in endodontic lesions associated with enrichment of pathogenic taxa more commonly associated with extra-oral infection and reflected those species previouslyfound to be ARB in amoebal/bacterial co-cultures [6,14,26,27], and those most commonly isolated from oral and non-oral mixed anaerobic infection, particularly of *Actinomyces sp.*, as well as *Bilophila sp*, *Porphyromonas sp.* and members of *Peptostreptococcus* spp. [28,29]. With wider reference to extra-oral infections, HACEK (*Haemophilus sp.*, *Aggregatibacter species*, *Cardiobacterium hominis*, *Eikenella corrodens*, and *Kingella sp.*)group members were enriched, which are linked to systemic disease in humansi [25]. Whilst classified within this study to family level only, *Cardiobacterium hominis* and *Eikenella corrodens* are commonly isolated in our laboratory from endodontic lesions, however, a significant endodontic contribution to systemic HACEK pathology remains to be demonstrated. Furthermore, a group formerly termed *Nutritionally Variant Streptococci* causative agents of endocarditis and bacteraemia, with higher mortality rates than other bacterial agents [30], have been described with endodontic and extraradicular infections [29,31], as well as in the present study with enrichment of *Granulitacella sp.* In lesions harbouring *E.gingivalis*. With the presence of *E. gingivalis* traits for resistance to phagocytosis (and extraradicaular manifestations), such as by selection for capsule formation (*Streptococcus sp.*) or filamentous growth (*Rothia sp.* or *Actinomyces sp.*). Supporting the present findings, these species reflect those observed free living and antibiotic treated free amoeba, found to be harbour similar *genera*, including *Enterococcus sp.*, *Rothia spp., Staphylococcus spp.* and *Streptococcus* spp. [6,12]. Regarding the diverse group of *streptococci*, the present study demonstrated an associated enrichment, but no strong conclusions could be drawn given methodological constraints and a single time point of sampling. In addition to the observation of conditioning capsule formation by amoeba, co-culture models have been proposed and validated to ascertain human virulence traits of Streptococcal capsules [9].

Together with previous observations that *E. gingivalis* may affect both the microbiome and the human host [15,16], our data support this enrichment of *genera* and species commonly associated with mixed, anaerobic infections. These are frequently delineated by *Lactobacillus*-depauperate states. and are associated with the outgrowth of facultative and strict anaerobes [32], reflected in the presence of many of these within a condensed clade indicative of recurrent infections in this study.

We demonstrated an amoebal enrichment of bacteria previously described within extra-radicular infections [33], and both the association of these classes of microbes with human infections, points to the fact that *amoebae* might be the motile carrier for these to the extra-radicular space, whilst many other, more common bacteria present in the canal, are not. It remains to be elucidated mechanistically how the presence of *amoeba* contribute to endodontic pathology, and if so, whether this occurs through stress responses to nutrient depletion in a mature community or through chemotaxis, as has been observed for *Entamoeba histolytica* in response to immune mediators, such as TNF-alpha, abundant around infected roots [34]. Further studies may explain the translocation and recruitment to specific sites within the oral environment, including associated bacteria.

Whilst the age-association is surprising, the observation of age-association of communities may have several confounders relating to endodontic lesions, including the time since lesion onset, some of the species identified in endodontic infection were atypical members of the previously described oral and endodontic microbiome. However, with acute dental abscesses, *S. aureus* has been reported in 0.7 to 15% of cases, with a higher recovery of coagulase-negative Staphylococci (4–65%), typically *S. epidermidis* [4,23,28,29], reported to occur more frequently in severe dental abscesses from younger patients and children [35,36]. The finding of an association with treatment signs, symptoms and stages, as well as age, merits further investigation, suggesting potentially separate disease entities, and may have ramifications for endodontic treatment.

Our data highlights that the interaction of bacteria and microorganisms derived from other kingdoms may result in conditioning and adaption leading to increased virulence within the host. Similar to strategies observed in human pathogens, studies in natural ecosystems of aquatic and terrestrial environments consistently demonstrated bacterial adaptation, such as changes in size and shape, motility, fimbria formation, capsule formation or toxin production to prevent or limit ingestion or digestion, taxa-specific differences in grazing patterns and food preferences on bacteria by free-living amoebae [6,10,11,13,17,37]. Taxa-specific features were enriched, including capsule, or those growing as filaments was observed, such as on *Actinomyces sp.* [6,7,8]. In addition to avoidance of phagocytosis, it has been suggested that *amoebae* represent an important reservoir for bacteria remaining viable within *amoeba*-derived specialised vacuoles [26], constituting parasitic or symbiotic relationships. Likened to Trojan horses, these organisms exploit amoebal adaptation, motility and display reduced sensitivity to antimicrobials and disinfectants, and serve as a ‘training ground’ of the bacteria, as ARB within amoeba share many common pathophysiologicalfeatures inside human macrophages [27,37], suggesting increased fitness and pathogenicity [38]. The presence of amoebal bacterial reservoir within the immuno-incompetent endodontic niche may contribute to the chronicity and exacerbation of chronic endodontic infection and explain the occasionally delayed failure of root canal treatments through re-emergence of infections, however further work is warranted to ascertain the clinical significance of amoebae in endodontic disease.

Even though the presence of *E. gingivalis* has been long implicated as disease-specific [15,16,39,40], little is known of the physiology or metabolic repertoire in terms of adaptation and virulence. Specific characterisation of the protists within this study was hampered further by the absence of any significant sequence data from this organism and difficulty in establishing these through axenic culture.

## 5. Conclusions

The finding, and the high prevalence, of protists within the setting of endodontic infections is relevant. The prevalence of sequence signatures in such a high number of samples suggests a role for amoebae in modulating endodontic pathology and might help understand this complex disease entity. Further work is warranted to understand these intra-kingdom interactions within disease processes and progression through conditioning and additional selective, as well as adaptive pressures.

These findings suggest that the conditioning pressures on bacterial communities, exerted by predatory protists within the endodontic infection setting, may ‘off-target’ increase human virulence by selecting for bacteria of increased resistance to protists, and possibly human phagocytosis. These findings suggest the need for novel targeting strategies and the recognition of protists as potential categorical modifiers in bacteria-driven human infections.

The importance of this study is to highlight the high prevalence of such modifiers in bacterially-driven infections and how insights from ecological studiees may help explain complex human infections.

## Figures and Tables

**Figure 1 jcm-09-03700-f001:**
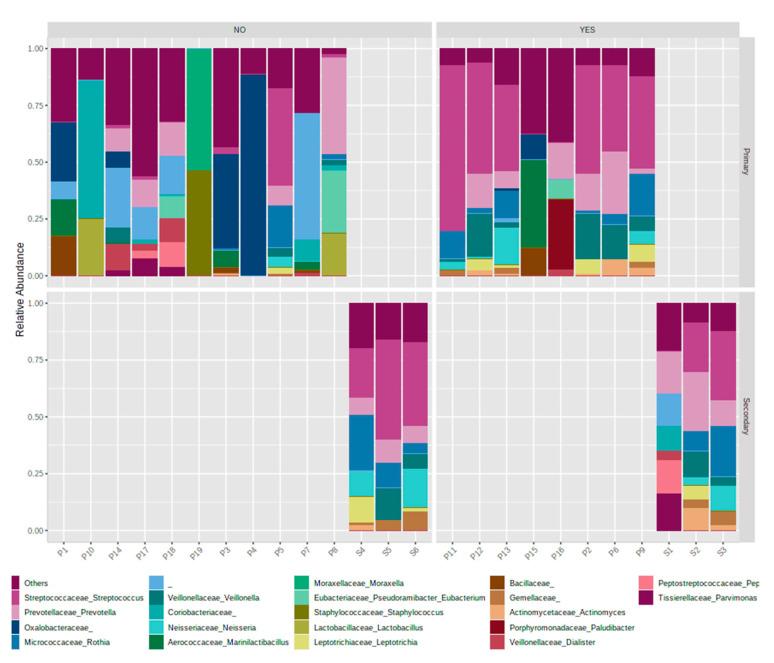
Taxonomic distribution and richness at *genus* level for samples across all endodontic infection samplesfor individual samples of primary (P1–19) and secondary cases (S1–6), delineated by Entamoeba status (YES/NO) and infection type (primary/secondary). *Entamoeba* positive cases displayed a reduced heterogeneity of genera present with predominance of taxa *Streptococcus sp.*, *Prevotella sp.* and *Rothia sp.* The diversity observed demonstrated similar predominant taxa between entamoeba-positive cases and secondary infections.

**Figure 2 jcm-09-03700-f002:**
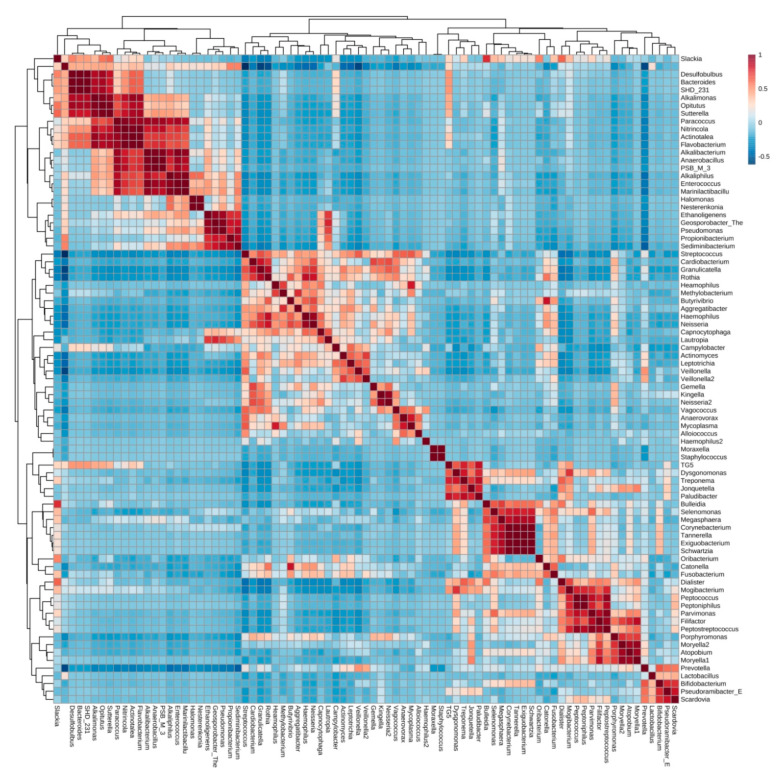
PearsonR correlated heatmap of bacterial *genera* in endodontic infections, highlighting distinct positive and negative correlations and categorical clusters between *genera*, defining the population composition and potential niche redundancies between closely related taxonomic groups. Scale represents correlation coefficients (+1 to −1). A high-resolution image of the present figure is provided in the online version of this manuscript.

**Figure 3 jcm-09-03700-f003:**
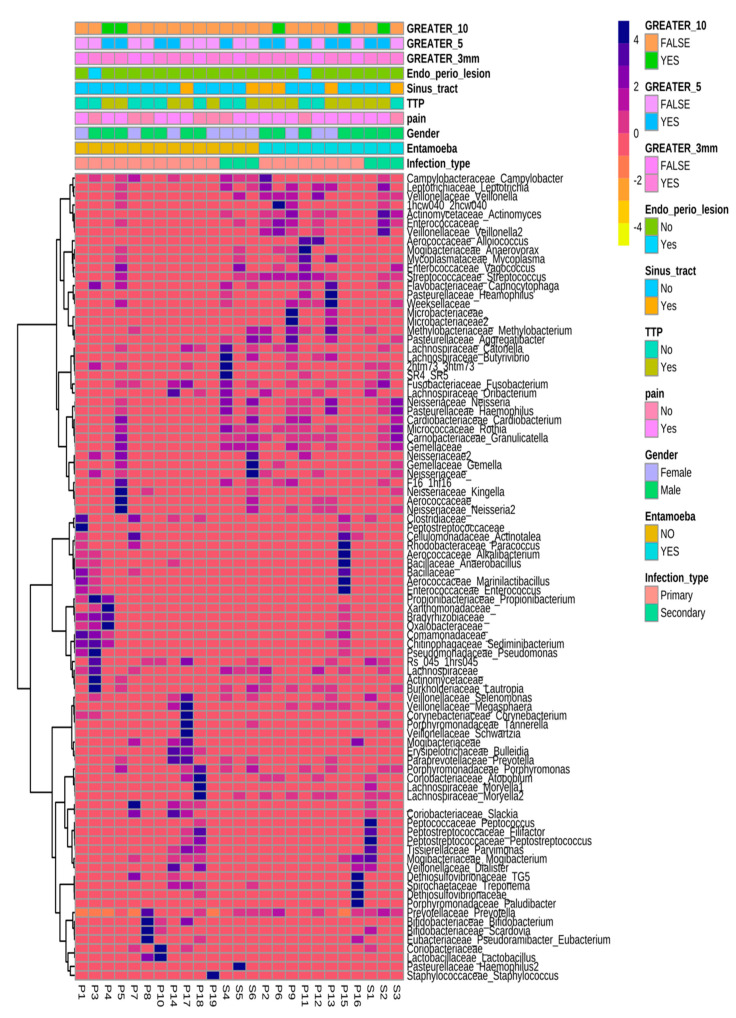
Ward clustered, genus-level heatmap aligned by sample identity and sample metadata by infection type and *entamoeba* status. Abundant clusters for secondary infections were noted in the first major cluster, with greater diversity and richness observed for primary infections at this taxonomic level. Greater x denoted lesion sizes (>3, >5, >10 mm, respectively), Tenderness to percussion (TTP). Scale represents relative log2 expression (−4 to 4). P, primary and S, secondary.

**Figure 4 jcm-09-03700-f004:**
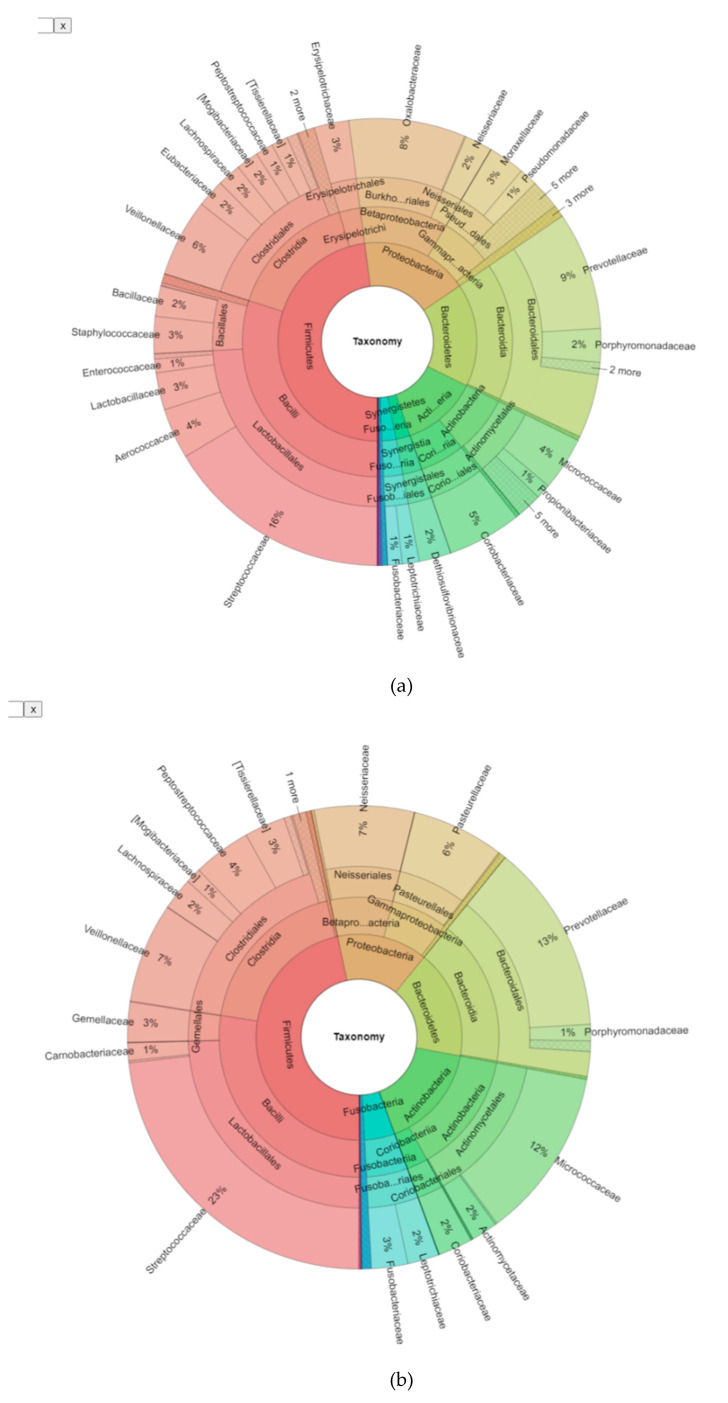
(**a**–**c**): Differential analysis of primary and secondary endodontic lesions. in primary (**a**) and secondary infections (**b**), demonstrating similar predominant *phyla* (*Firmicutes* and *Bacteroidetes*), with shifts resulting in decreased abundance of *Proteobacteria* and *Synergistetes*, increased abundance of *Actinobacteria*, *Fusobacteria* and *Saccharibacteria* in the transition to secondary infections; (**c**) Linear Discriminant Analysis (LDA) Effect Size (LefSe) for OTUs selectively enriched for both groups, such as *Rothia mucilaginosa* (out013), for secondary infections and (non-aureus) *Staphylococcus* sp. (OTU050), *Kingella* sp. (OTU121) and *Bifidobacterium* sp. (OTU018) for primary infections., out OTU identities are provided in Appendix A.

**Figure 5 jcm-09-03700-f005:**
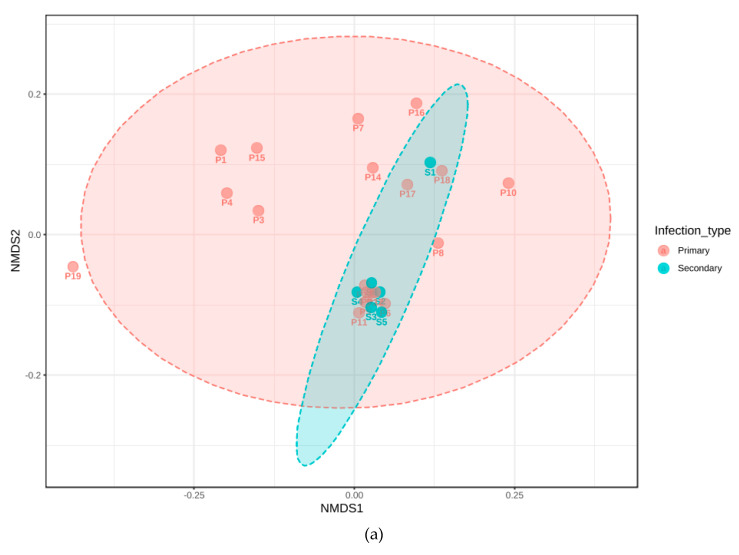
(**a**–**e**): Multidimensional scaling analysisesof microbiomal diversity, correlated to clinical parameters, displaying the effect on dispersion by sample number at *genus* level for metadata analysed, based on infection types (**a**), presence of sinus tracts (**b**), pain (**c**), tenderness to percussion (**d**) and gender (**e**). The cluster analysis demonstrated significant clustering by PERMDISP for infection type (*p* < 0.01) and the presence of sinus tract (*p* < 0.01). A more condensed clustering, covering secondary cases was noted. NMDS—non-metric multidimensional scaling.

**Figure 6 jcm-09-03700-f006:**
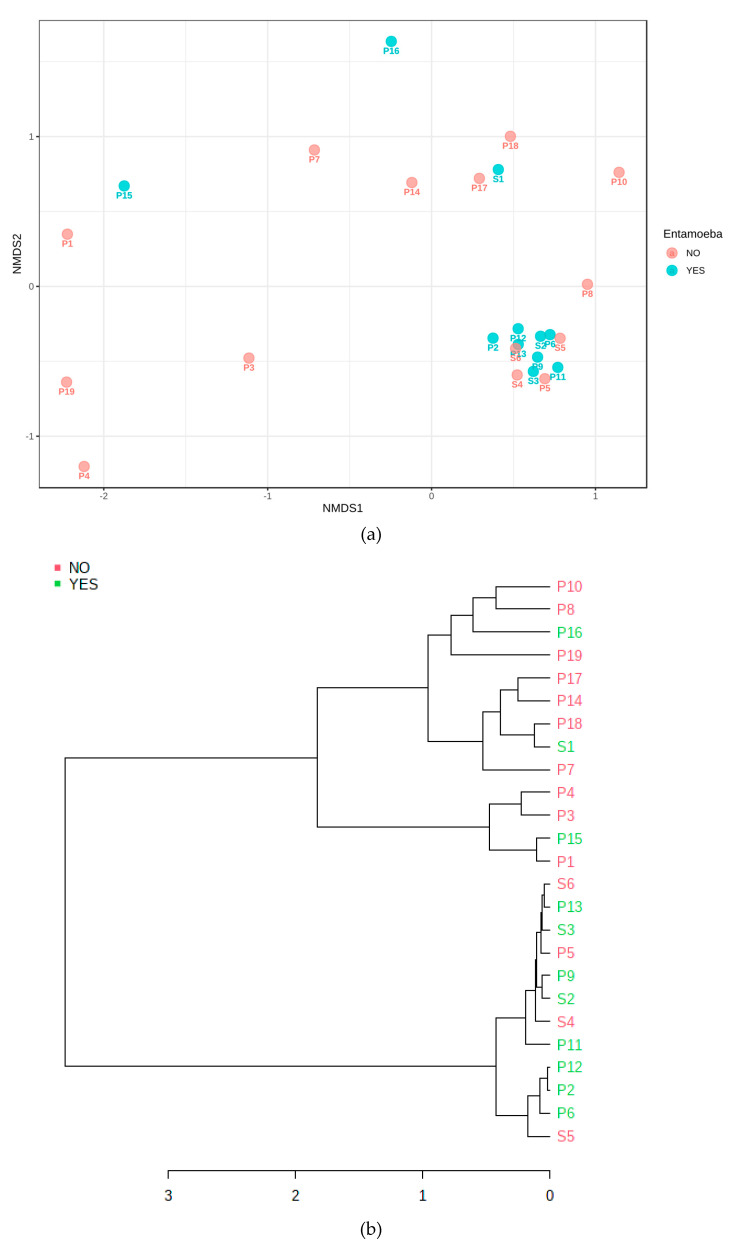
(**a**–**c**): Correlation of primary (P1–P19) versus secondary (S1–S6) infections and Entamoeba status (Yes/No). (**a**) Multidimensional NMDS plot and (**b**) Jenson-Shannon clustered, phylogenetic chart displaying a core, phylogenetically dense and related cluster of both secondary infections and Entamoeba-positive status. The LefSe enrichment (**c**) of genera demonstrated significant differential enrichment based on Entamoeba status..

**Table 1 jcm-09-03700-t001:** Overview of clinical metadata

**Infection type**	Primary/Secondary %	76:2
**Gender**	Gender Ratio M:F %	52:48
**Age**	Age (Years)	42.1 (SD 14.2)
**Symptoms**	Pain (Y/N) %	83:17
**Signs**	Tenderness to percussion (Y/N) %	56:44
**Apical radiolucency extent**	Largest extent of radiolucency (mm)	average 6.7 mm (0–25.4 mm)
**Distribution of large radiolucencies**	Largest extent of radiolucency diameter > 5 mm (Y/N) %	52:48
**Fistula**	Sinus tract present (Y/N) %	20:80

Infection types (%), demographic information and signs/symptoms of patients recruited. Infections types related to primary treatment, i.e., treatment naive teeth or those previously having failed root canal treatment (secondary).

**Table 2 jcm-09-03700-t002:** The 25 most commonly identified *genera* and species across all endodontic samples examined, ordered by prevalence. Detailed taxonomic contributions are presented in Appendix A.

Genera (In Order of Abundance)	Species (In Order of Abundance)
*Streptococcus, Prevotella, Rothia, Veillonella, Neisseria, Marinilactobacillus, Lactobacillus, Staphylococcus, Moraxella, Solobacterium (formerly Bulleidia), Pseudoramibacter/Eubacterium, Peptostreptococcus, Fusobacterium, Leptotrichia, Parvimonas, Dialister, Haemophilus, Paludibacter, Atopobium, Pseudomonas, Actinomyces, Porphyromonas, Enterococcus, Corynebacterium, TG5 and Granulicatella*	*Rothia mucilaginosa, Prevotella melaninogenica, Veillonella dispar, Neisseria subflava, Marinilactobacillus psychrotolerans, Staphylococcus aureus, Haemophilus parainfluenzae, Prevotella nigrescens, Prevotella tannerae, Prevotella pallens, Propionibacterium acnes, Lactobacillus zeae, Rothia dentocariosa, Rothia aeria, Lactobacillus reuteri, Streptococcus anginosus, Porphyromonas endodontalis, Aggregatibacter segnis, Corynebacterium durum, Prevotella nanceiensis, Treponema socranskii, Haemophilus parahaemolyticus, Alkaliphilus transvaalensis, Veillonella parvula, Haemophilus influenzae*

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
