# Peer review of "Amoebae in Chronic, Polymicrobial Endodontic Infections Are Associated with Altered Microbial Communities of Increased Virulence"

_jcm, 2020, doi:10.3390/jcm9113700_

Round 1

Reviewer 1 Report

Even though readability and English language of the manuscript has been improved, many changes and corrections, especially regarding the figures, unfortunately have not been performed.

Even though the scientific work is good and the experiments were all carried out well and the applied statistical methods are correct, the quality of a manuscript is not only determined by the scientific work but also by its presentation which should consist of comprehensible and intelligible figures and text which conveys the findings in a clear way. This has not been achieved, although the manuscript has been revised.

Methods

Please provide Primer sequences and PCR s as well as concentrations of the PCR components for the fungal ITS gene amplification as well as the E. gingivalis detection.

Results

Table1: the font is improved, but the formatting shifted, please correct

L129: could you please state the results of the mock community in detail? Which taxa were part of the mock community and which taxa were detected? Were they detected in the right abundance?

Figure 1: the font size is not changed, readability of the Figures is not achieved. Please truly enlarge font size of the figure caption. The nicest figure is of no use if you can’t even read the captions.

Fig.3: In some cases the colours used for the sample data cannot be discriminated very well, e.g. shades of green for “pain”. Please change. Also, values -4 to 4 are not explained. Please add description. This has not been done.

Fig. 5 and L243: a)-f) is inserted in the figure but the figure legend only explains a) and b) and b) corresponds to f) in the Figure. Also, the p value in the Figure legend states PERMDISP) (p<0.001) but in L243 the p value is 0.024. Which one is correct?

Fig. 6: Figure 6 is not referred to in the text of the results section or explained there. Please change. This has not been done in the revised text.

L 165-167 It is not clear to me, how Figure 2 shows that some primary cases are clustering more closely with secondary cases. Please explain Fig. 2 a bit better. This has not been done, I cannot find a more detailed explanation of Fig.2.

Discussion:

In the discussion an association of E. gingivalis with secondary endodontic infections is stated. I find it very hard to reproduce this on the basis of the figures of the results especially since Fig 6 is mentioned nowhere in the results. Also, the respective taxa that are mentioned in L365-373 of the discussion are not identifiable in the Figures of the results section. Please change and correct.  Also, you keep adding results in the discussion section that you haven’t mentioned in the results section. Please limit the presentation of results to the results section and interpret these in the discussion section.

L.339-342: Sentence is unclear, please rephrase. This has not been done.

L.365-371: should this not reflect what is stated in L. 243-254 where you stated the associated species with E. gingivalis? This point has not been clarified in the Revision.

  1. 357: where are the Nutritionally Variant streptococci shown in the results section? This has not been made clear in the results section.
  2. 387-395: sentence long and incomprehensible. This has not been changed.

Minor changes:

  1. 180-182: syntax of this sentence is incorrect, please rephrase. Not corrected yet

L.335: as well as Not corrected yet

Author Response

We gratefully acknowledge the esteemed reviewers’ thoughtful comments and detailed critique of our manuscript.

In light of the comments, we have carried out an extensive revision of the paper, and have replotted the images to provide adequate resolution. In the addition to higher quality images in-line within the paper, ee have submitted electronic, high-resolution images as part  of this present submission. We have amended the images to improve readability, and have resolved (where appropriate) to genus level to aid comprehension. We have carried out further revision of the text to improve scientific quality and have edited sections as requested to support the key results and discussion.

For clarity, we have enclosed a track changed version and an untracked version and have provided replied with reference to the untracked version below.

Even though readability and English language of the manuscript has been improved, many changes and corrections, especially regarding the figures, unfortunately have not been performed. Even though the scientific work is good and the experiments were all carried out well and the applied statistical methods are correct, the quality of a manuscript is not only determined by the scientific work but also by its presentation which should consist of comprehensible and intelligible figures and text which conveys the findings in a clear way. This has not been achieved, although the manuscript has been revised.

We agree with the above points and the requested modifications have been carried out

Methods

Please provide Primer sequences and PCR s as well as concentrations of the PCR components for the fungal ITS gene amplification as well as the E. gingivalis detection.

The primer sequences and conditions were added in detail and can be found in Appendix section A, and references made to this section in the methodology.

Results

Table1: the font is improved, but the formatting shifted, please correct

This has been corrected.

L129: could you please state the results of the mock community in detail? Which taxa were part of the mock community and which taxa were detected? Were they detected in the right abundance?

The wording has been changed to non-template control, we apologise for the confusion caused. The controls had very low levels of template and failed NGS QC in terms of quantity and quality, but in a few instances, where these were eliminated from further analysis.

Figure 1: the font size is not changed, readability of the Figures is not achieved. Please truly enlarge font size of the figure caption. The nicest figure is of no use if you can’t even read the captions.

The images have been replotted and high resolution images provided. Larger, higher resolution images have been provided in the body of the manuscript. Where increased font size was not possible, such as for Figure 2, we have reanalyzed the data to provide genus-level information, which improves readability and improves comprehension over the full OTU comparison previously carried out. We are grateful for these helpful comments.

Fig.3: In some cases the colours used for the sample data cannot be discriminated very well, e.g. shades of green for “pain”. Please change. Also, values -4 to 4 are not explained. Please add description. This has not been done.

We have replotted the images to provide better contrast, details and resolution. A higher resolution and detail has been provided for the high resolution submission. Also the color patterns have been changed to increase discrimination.

Fig. 5 and L243: a)-f) is inserted in the figure but the figure legend only explains a) and b) and b) corresponds to f) in the Figure. Also, the p value in the Figure legend states PERMDISP) (p<0.001) but in L243 the p value is 0.024. Which one is correct?

We apologise for the oversight. The p values related to OTU or genus level and we have used the genus level analysis to provide consistency., we have amended these to reflect the p=0.024 value applicable/

Fig. 6: Figure 6 is not referred to in the text of the results section or explained there. Please change. This has not been done in the revised text.

The reference to the figure has been added to the text as requested

L 165-167 It is not clear to me, how Figure 2 shows that some primary cases are clustering more closely with secondary cases. Please explain Fig. 2 a bit better. This has not been done, I cannot find a more detailed explanation of Fig.2.

A further explanation to Fig. 2 has been added within the text.

Discussion:

In the discussion an association of E. gingivalis with secondary endodontic infections is stated. I find it very hard to reproduce this on the basis of the figures of the results especially since Fig 6 is mentioned nowhere in the results. Also, the respective taxa that are mentioned in L365-373 of the discussion are not identifiable in the Figures of the results section. Please change and correct.  Also, you keep adding results in the discussion section that you haven’t mentioned in the results section. Please limit the presentation of results to the results section and interpret these in the discussion section.

We tuned down the sentences which could be interpreted as results to the level of hypothesis relevant to the discussion. Taxa level is now level is now explicited.

L.339-342: Sentence is unclear, please rephrase. This has not been done.

The sentence has been modified, to provide a clearer undertanding of the hypothesis

L.365-371: should this not reflect what is stated in L. 243-254 where you stated the associated species with E. gingivalis? This point has not been clarified in the Revision.

  1. 357: where are the Nutritionally Variant streptococci shown in the results section? This has not been made clear in the results section.

The variants have been specified and the NGS used in the literature

  1. 387-395: sentence long and incomprehensible. This has not been changed.

This has now been modified to increase readability

Reviewer 2 Report

Thank you for submitting the review report. The revised manuscript was well responded point by point according to the reviewer's comments.

Author Response

Thank you very much, we appreciate

Round 2

Reviewer 1 Report

Review JCM jcm-921665

In this revised version, the readability and comprehensibility of the figures has been improved significantly. Also, overall clarity concerning the main findings and their interpretations has been achieved by editing the text.

This manuscript is a resubmission of an earlier submission. The following is a list of the peer review reports and author responses from that submission.

Round 1

Reviewer 1 Report

General:

This manuscript is well-written and easy to follow. The authors showed that amoebae were associated with altered microbial communities of increased virulence in endodontic infection, which is clinically important to both endodontist or mycologists.

Specific:

1.Figures 1, 4, 5, & 6 have sub-pictures and add the alphabet (a, b,c, d....) below each picture for discrimination.

2. Please read carefully the author's guidelines because the formats of Tables 1 & 2 are wrong: do not need vertical & horizontal lines.

3. There are so many typos in references section: the name of organisms should be written in italic in the titles of published journal. Also in #16 reference, only first alphabet of the first word must have a capital letter.

4. This manuscript mainly described the microbial diversity and the richness from the primary and secondary endodontic infections. For endodontist, it is very interesting for the authors to describe what clinical signs and symptoms are associated with the existence of amoebae in Discussion section if the authors can.

5. Please make compact the Conclusions since the paragraphs are too lengthy.

Author Response

The authors gratefully acknowledge the reviewers’ helpful comments and constructive critique of our manuscript. We agree with the points raised and have amended the manuscript to reflect these comments.  

 JCM jcm-921665 

General: 

This manuscript is well-written and easy to follow. The authors showed that amoebae were associated with altered microbial communities of increased virulence in endodontic infection, which is clinically important to both endodontist or mycologists. 

Specific:

1.Figures 1, 4, 5, & 6 have sub-pictures and add the alphabet (a, 
b,c, d....) below each picture for discrimination. 

We are grateful for the comment and have addressed the editing. 

2. Please read carefully the author's guidelines because the formats of Tables 1 & 2 are wrong: do not need vertical & horizontal lines.

His has been amended, as advised 

3. There are so many typos in references section: the name of organisms should be written in italic in the titles of published journal. Also in #16 reference, only first alphabet of the first word must have a capital letter. 

This has been amended as requested. 

4. This manuscript mainly described the microbial diversity and the richness from the primary and secondary endodontic infections. For endodontist, it is very interesting for the authors to describe what clinical signs and symptoms are associated with the existence of amoebae in Discussion section if the authors can. 

WE have addressed this point, including shortcomings of the present study precluding the inference of a direct, causation rather than association. 

5. Please make compact the Conclusions since the paragraphs are too lengthy.

This has been amended as requested. 

Reviewer 2 Report

Review JCM jcm-921665

Investigating the microbiota involved in the development of endodontic infection with regard to amoeba is of interest since information on this species’ involvement in endodontic infection is scarce. So the rationale of the study is justified. However, even though the study seems to have yielded some interesting results, the way in which these are formulated and presented in the results section and referred to in the discussion section is done in such an unclear and confused way that it fails to convey the main points in a clear and concise way. Many sentences are too long and incomprehensible and also flawed regarding their syntax. In addition there is often vague argumentation and confusing associations that are not congruent between results and discussion section. Therefore most of the results section and especially the discussion section need to be rewritten in a manner that conveys the results in a clearly comprehensible and replicable way.

Also incorrect syntax and false comma placement renders many sentences very unclear, therefore English language correction is necessary.

Results, Discussion

L 165-167 It is not clear to me, how Figure 2 shows that some primary cases are clustering more closely with secondary cases. Please explain Fig. 2 a bit better.

L.183: Do you mean Appendix B figures by “supplementary Data 1”? It would help to use consistent denotations of the respective figures. Also please number the Appendix Figures

L.229-231: Sentence is unclear and conveys a different meaning than sentence in L.298. Please rephrase according to meaning of sentence in L.298-299 (legend of Fig. 5)

L.258: This is an interpretation of the results and should be placed and discussed in the discussion section.

Discussion:

In the discussion an association of E. gingivalis with secondary endodontic infections is stated. I find it very hard to reproduce this on the basis of the figures of the results especially since Fig 6 is mentioned nowhere in the results. Also, the respective taxa that are mentioned in L365-373 of the discussion are not identifiable in the Figures of the results section. Please change and correct.  Also, you keep adding results in the discussion section that you haven’t mentioned in the results section. Please limit the presentation of results to the results section and interpret these in the discussion section.

L.328-33: Syntax is unclear, please rephrase correctly.

L.339-342: Sentence is unclear, please rephrase.

L.355: Whilst this study was unable to demonstrate clear association between the onset of symptomatic onset? Sentence is unclear, please rephrase (association between what and what?)

L.356: How can you state a “condensed diversity” in secondary infections, although you haven’t stated any results about the diversity parameters comparing primary and secondary infections in the results section?

L.358: What is the “chronic cluster”?

L.365-371: should this not reflect what is stated in L. 243-254 where you stated the associated species with E. gingivalis?

L.357: where are the Nutritionally Variant streptococci shown in the results section?

L.384-387: Sentence is unclear, please rephrase.

L.387-395: sentence long and incomprehensible

Tables:

Table 1: please use a larger font size

Figures:

Fig. 1: the small font size makes the legends hard to read; Labels a-d are present in the legend but missing in the Figure; the meaning of the values (-0.5 -1.0) in the legend of Figure 1d is missing.

Fig. 2 Red and blue values are not mentioned/explained in Figure legend. Please correct.

Fig.3: In some cases the colours used for the sample data cannot be discriminated very well, e.g. shades of green for “pain”. Please change. Also, values -4 to 4 are not explained. Please add description.

Fig. 4: a,b,c,d are present in legend but missing in actual Figure. Please correct. Pie charts are not well comparable since different colours stand for different taxa. It would be improved if same colours could be used for same taxa. Also, for Figure 4d and 4e please add the taxonomic assignment (e.g. genus or species) for the respective OTUs in the plot legend.

Fig. 5: Please add the taxonomic assignment for the resp. OTUs in the plot legend of 5d.

Fig. 6: Figure 6 is not referred to in the text of the results section or explained there. Please change.

Minor changes:

L133: rephrase sentence and avoid doubling

L.180-182: syntax of this sentence is incorrect, please rephrase

L.227 and L.554: Positive associations of reads and radiolucency size size; delete “size”

L.335: as well as

L.359: even though instead of even the

Author Response

The authors gratefully acknowledge the reviewers’ helpful comments and constructive critique of our manuscript. We agree with the points raised and have amended the manuscript to reflect these comments.  

 JCM jcm-921665 

General: 

This manuscript is well-written and easy to follow. The authors showed that amoebae were associated with altered microbial communities of increased virulence in endodontic infection, which is clinically important to both endodontist or mycologists. 

Specific: 

1.Figures 1, 4, 5, & 6 have sub-pictures and add the alphabet (a, b,c, d....) below each picture for discrimination. 

We are grateful for the comment and have addressed the editing. 

  1. Please read carefully the author's guidelines because the formats of Tables 1 & 2 are wrong: do not need vertical & horizontal lines.

His has been amended, as advised 

  1. There are somany typos in references section: the name of organisms should be written in italic in the titles of published journal.Also in #16 reference, only first alphabet of the first word must have a capital letter. 

This has been amended as requested. 

  1. This manuscript mainly described the microbial diversity and the richness from the primary and secondary endodontic infections. For endodontist, it is very interesting for the authors to describe what clinical signs and symptoms are associated with the existence of amoebae in Discussion section if the authors can. 

WE have addressed this point, including shortcomings of the present study precluding the inference of a direct, causation rather than association. 

  1. Please make compact the Conclusions since the paragraphs are too lengthy.

This has been amended as requested. 

REVIEWER 2 

Investigating the microbiota involved in the development of endodontic infection with regard to amoeba is of interest since information on this species’ involvement in endodontic infection is scarce. So the rationale of the study is justified. However, even though the study seems to have yielded some interesting results, the way in which these are formulated and presented in the results section and referred to in the discussion section is done in such an unclear and confused way that it fails to convey the main points in a clear and concise way. Many sentences are too long and incomprehensible and also flawed regarding their syntax. In addition there is often vague argumentation and confusing associations that are not congruent between results and discussion section.  Therefore most of the results section and especially the discussion section need to be rewritten in a manner that conveys the results in a clearly comprehensible and replicable way. 

We are very grateful to the esteemed reviewer for these very helpful comments. The discussion section has been extensively edited to address these helpful comments. We have amended statements highlighting shortcomings and have made the above clearer. 

Also incorrect syntax and false comma placement renders many sentences very unclear, therefore English language correction is necessary. 

This has been carried out by native speaker 

Results, Discussion 

L 165-167 It is not clear to me, how Figure 2 shows that some primary cases are clustering more closely with secondary cases. Please explain Fig. 2 a bit better. 

This point has been addressed in the accompanying text to reflect the reduced diversity of secondary cases. 

L.183: Do you mean Appendix B figures by “supplementary Data 1”? 

The references to appendices have been amended to reflect this throughout.  

 It would help to use consistent denotations of the respective figures. Also please number the Appendix Figures 

The images have been amended as requested.  

L.229-231: Sentence is unclear and conveys a different meaning than sentence in L.298. Please rephrase according to meaning of sentence in L.298-299 (legend of Fig. 5) 

This has been amended.  

L.258: This is an interpretation of the results and should be placed and discussed in the discussion section. 

This has been amended accordingly. 

Discussion: 

In the discussion an association of E. gingivalis with secondary endodontic infections is stated. I find it very hard to reproduce this on the basis of the figures of the results especially since Fig 6 is mentioned nowhere in the results. Also, the respective taxa that are mentioned in L365-373 of the discussion are not identifiable in the Figures of the results section. Please change and correct. Also, you keep adding results in the discussion section that you haven’t mentioned in the results section. Please limit the presentation of results to the results section and interpret these in the discussion section. 

We appreciate the esteemed reviewer’s comments and have heavily edited the discussion section to address the concerns raised. Specifically, findings unsupported by results presented have been removed.  

L.328-33: Syntax is unclear, please rephrase correctly. 

This has been amended as requested. 

L.339-342: Sentence is unclear, please rephrase. 

This has been amended as requested. 

L.355: Whilst this study was unable to demonstrate clear association between the onset of symptomatic onset? Sentence is unclear, please rephrase (association between what and what?) 

The section has been amended accordingly. 

L.356: How can you state a “condensed diversity” in secondary infections, although you haven’t stated any results about the diversity parameters comparing primary and secondary infections in the results section? 

The wording has been made clearer and more detail has been provided. 

L.358: What is the “chronic cluster”? 

This has been amended as requested to clearly describe the cluster in question.  

L.365-371: should this not reflect what is stated in L. 243-254 where you stated the associated species with E. gingivalis? 

This has been addressed to describe both methods utilised. 

L.357: where are the Nutritionally Variant streptococci shown in the results section? 

This will be rewritten to state common, systemic organisms and remove any inference of significance to endodontics 

L.384-387: Sentence is unclear, please rephrase. 
This has been amended as requested. 

L.387-395: sentence long and incomprehensible 
this section has been rewritten as requested. 

Tables: 

Table 1: please use a larger font size 

This has been amended as requested. 

Figures: 

Fig. 1: the small font size makes the legends hard to read; Labels a-d are present in the legend but missing in the Figure; the meaning of the values (-0.5 -1.0) in the legend of Figure 1d is missing. 

Addressed to explain correlation coefficients and legends provided. 

Fig. 2 Red and blue values are not mentioned/explained in Figure legend. Please correct. 
This has now been amended to address the sample points in the principal component plots. 

Fig.3: In some cases the colours used for the sample data cannot be discriminated very well, e.g. shades of green for “pain”. Please change. Also, values -4 to 4 are not explained. Please add description. 

Addressed to make the samples consistent across groups. 

Fig. 4: a,b,c,d are present in legend but missing in actual Figure. Please correct. Pie charts are not well comparable since different colours stand for different taxa. It would be improved if same colours could be used for same taxa. Also, for Figure 4d and 4e please add the taxonomic assignment (e.g. genus or species) for the respective OTUs in the plot legend. 
Addressed and clear reference to OTU table provided 

Fig. 5: Please add the taxonomic assignment for the resp. OTUs in the plot legend of 5d. 

We have included references to the OTU in every instance. 

Fig. 6: Figure 6 is not referred to in the text of the results section or explained there. Please change. 
Addressed within the body of the text 

Minor changes: 

L133: rephrase sentence and avoid doubling 

This has been amended. 

L.180-182: syntax of this sentence is incorrect, please rephrase 

The sentence has been rewritten to reflect the esteemed reviewer’s comments. 

L.227 and L.554: Positive associations of reads and radiolucency size size; delete “size” 

This has been amended as requested. 

L.335: as well as 

This has been amended as requested. 

L.359: even though instead of even the 

This has been amended as requested.